# Affibody Modified G-quadruplex DNA Micelles Incorporating Polymeric 5-Fluorodeoxyuridine for Targeted Delivery of Curcumin to Enhance Synergetic Therapy of HER2 Positive Gastric Cancer

**DOI:** 10.3390/nano12040696

**Published:** 2022-02-19

**Authors:** Chao Zhang, Shuangqing Fu, Fanghua Zhang, Mengnan Han, Xuming Wang, Jie Du, Honglei Zhang, Wei Li

**Affiliations:** 1College of Chemistry and Environmental Science, Key Laboratory of Chemical Biology of Hebei Province-Laboratory of Medicinal Chemistry and Molecular Diagnosis of the Ministry of Education, Institute of Life Science and Green Development, Hebei University, Baoding 071002, China; zhangchao_abcd@163.com (C.Z.); fsqlv123@126.com (S.F.); fang411335@163.com (F.Z.); mn2659290716@hotmail.com (M.H.); wangxuming9588@163.com (X.W.); dujie2008@126.com (J.D.); 2Department of Life Science, Hengshui University, Hengshui 053000, China

**Keywords:** G-quadruplex, DNA micelles, affibody, 5-fluorodeoxyuridine, curcumin, synergetic therapy

## Abstract

Combination chemotherapy is emerging as an important strategy for cancer treatment with decreased side effects. However, chemotherapeutic drugs with different solubility are not easy to realize co-delivery in traditional nanocarriers. Herein, an affibody modified G-quadruplex DNA micellar prodrug (affi-F/GQs) of hydrophilic 5-fluorodeoxyuridine (FUdR) by integrating polymeric FUdRs into DNA strands is developed for the first time. To achieve synergistic efficacy with hydrophobic drugs, curcumin (Cur) is co-loaded into affi-F/GQs micelles to prepare the dual drug-loaded DNA micelles (Cur@affi-F/GQs), in which affibody is employed as a targeting moiety to facilitate HER2 receptor-mediated uptake. Cur@affi-F/GQs have a small size of approximately 130 nm and exhibit excellent stability. The system co-delivers FUdR and Cur in a ratiometric manner, and the drug loading rates are 21.1% and 5.6%, respectively. Compared with the physical combination of FUdR and Cur, Cur@affi-F/GQs show higher cytotoxicity and greater synergistic effect on HER2 positive gastric cancer N87 cells. Surprisingly, Cur@affi-F/GQs significantly enhance the expression and activity of apoptosis-associated proteins in Bcl-2/Bax-caspase 8, 9-caspase 3 apoptotic pathway, which is the main factor in the death of tumor cells induced by FUdR. Overall, this nanoencapsulation is a promising candidate for the targeted co-delivery of drugs with significant differences in solubility.

## 1. Introduction

Combination chemotherapy is widely applied in the clinical treatment of cancer due to the complexity and high variability of tumors [1,2,3]. Two (or more) drugs against different disease pathways in the cancer process are employed to increase the chances of curing cancer, while the traditional single drug administration may be insufficient to achieve tumor recession in clinics [4]. The combination of chemotherapeutic drugs allows lower drug dose, reducing the cytotoxic effect and improving the efficacy, and so it provides a strategy worthy of in-depth study for cancer treatment [5,6]. However, the simple cocktail administration of multiple drugs often leads to uncertainty in cancer treatment due to the inconsistent pharmacokinetics and dissimilar biodistributions of different drugs, particularly drugs with strongly different aqueous solubility (i.e., hydrophobic and hydrophilic) [7,8]. Given the limitations in combination chemotherapy, nanocarriers have become powerful vehicles for the co-delivery of different drugs [9,10,11,12,13]. In the last few decades, nanocarriers of different materials, such as liposomes [14,15], micelles [16,17] and mesoporous silica [18], have been developed to co-deliver hydrophobic/hydrophilic drugs. However, many of these encapsulate drugs noncovalently, which may lead to inaccurate drug loading and inevitably premature drug leakage during blood circulation [19]. In addition, the physicochemical variability, biosafety and manufacturing complexity of the aforementioned nanomaterials make their clinical transformation extremely challenging [20]. Therefore, it is of the utmost urgency to devise a novel drug delivery system with precise drug loading, excellent biocompatibility, stability and biosafety to facilitate combination chemotherapy, particularly the co-delivery system of hydrophobic and hydrophilic therapeutics.

With the rapid development of DNA nanotechnology, DNA-based nanoparticles as drug nanocarriers have been an exciting field attracting lots of interest [21,22]. DNA as a biosafety material has excellent biocompatibility and low cytotoxicity, making it more suitable for biomedical applications. The specified DNA strands can be easily synthesized by commercial solid-phase synthesis technology to form nanoparticles with various structures through complementary pairing. In recent years, the supramolecular DNA micelles composed of novel amphiphilic DNA nanomaterial have been successfully developed [23]. A DNA micelle has a distinct core-shell structure. The hydrophilic shell consists of DNA strands and the hydrophobic core is formed by synthetic polymer (polypropylene oxide (PPO) [24], poly (D, L-lactic-co-glycolic acid) (PLGA) [25]) or natural lipid (phospholipid [26], cholesterol [27]). This structure is beneficial for encapsulating hydrophobic drugs and improving their bioavailability. Compared with traditional copolymer micelles, the preparation of DNA micelles is simple and fast, and the composition and structure can be precisely controlled. They also have the ability of molecular recognition, information storage and transfer, which have been gradually applied to drug delivery [24], gene therapy [28] and biosensors [29,30]. However, the spontaneous interaction of micelles and protein components in serum in vivo makes their structure unstable, which promotes dissociation of the micelles into monomers, thus leading to the release of drugs in the micelles in advance [31,32]. To circumvent the above problems, some researchers use tetramolecular G-quadruplex as adhesive to intermolecularly connect the outer shells of DNA micelles together and more compact and stable G-quadruplex-locked DNA micelles are obtained [33,34]. Tetramolecular G-quadruplex is a non-canonical nucleic acid supramolecular nanostructure formed by four chemically synthesized guanine-rich oligonucleotides through a Hoogsteen hydrogen bond and π-π stacking (Figure 1D) [35]. DNA micelles with tetramolecular G-quadruplex structure become more stable through DNA strand interaction and hydrophobic driving force, which makes the four amphiphilic DNA monomers closely bind to effectively prevent the destruction of serum albumin [36,37]. Although there are many reports on the design, assembly and stability of G-quadruplex DNA micelles [33,34,36,37], little research has been conducted on G-quadruplex DNA micelles as drug delivery. Thus, G-quadruplex DNA micelles, as stable drug carriers, are considered to have great potential in cancer therapy.

In addition to the tight and stable micelle structure, it is also very pivotal to confer G-quadruplex DNA micelles active targeting, which can effectively deliver therapeutic drugs to specific tumor sites and enhance treatment efficacy [38,39]. demonstrate that affibody molecule is an excellent target ligand that has the advantages of low molecular weight (7 kDa), stable structure and function, high selectivity and affinity [40,41,42]. In this regard, affibody as a targeting moiety will be employed to develop a variety of drug delivery systems for the treatment of HER2-positive tumors [43,44,45].

In this study, 5-fluorodeoxyuridine (FUdR, metabolite of 5-Fluorouracil) and curcumin (Cur) were chosen as a model hydrophobic drug and hydrophilic drug due to their good synergistic effect in cancer treatment [46,47,48]. 5-Fluorouracil (5-FU), as the main chemotherapeutic drug for terminal gastrointestinal cancer, can inhibit the synthesis of deoxythymidine monophosphate by covalent binding with thymidylate synthase (TS). It also restrains cell division by incorporating DNA or RNA, thus causing cell apoptosis [49]. However, owing to its short half-life (about 8–20 minutes) and poor selectivity, 5-FU is prone to serious adverse reactions and drug resistance, resulting in the unsatisfactory effect of chemotherapy. Therefore, 5-FU is commonly used in combination with other drugs in clinical trials to enhance the efficacy and reduce side effects. Cur, as a polyphenolic compound extracted from Curcuma longa Linn., has almost no toxicity [50] and can sensitize cancer cells to 5-FU by blocking the drug efflux and negatively regulating the related factors and genes expression [51,52]. In our previous articles, FUdRs were individually delivered or co-delivered accurately to the tumor site by a DNA tetrahedron, [43] DNA-RALA polymer [44] and DNA-gold nanoparticle [45]. Here, an affibody modified G-quadruplex DNA micelle co-loaded with FUdR and Cur (Cur@affi-F/GQs) were designed and prepared for the targeted combination treatment of HER2 -positive gastric carcinoma cell. The affibody-FUdR-G-quadruplex DNA micellar prodrug (affi-F/GQs) were synthesized as drug delivery systems for the first time. In addition, the characterization of affi-F/GQs, such as morphology, particle size and stability, were conducted. The affibody-mediated specific binding was also evaluated in gastric carcinoma cancer cells with different levels of HER2 expression. According to the characteristic of micelle, Cur was encapsulated into the hydrophobic core of affi-F/GQs by physical entrapment and Cur@affi-F/GQs co-loaded with FUdR and Cur were obtained. Then, the cytotoxicity of Cur@affi-F/GQs and synergistic chemotherapy efficacy of FUdR and Cur were assessed systemically. Moreover, the apoptosis-related proteins were evaluated to research the impact mechanism of Cur@affi-F/GQs. The results revealed that affibody-G-quadruplex DNA micelles may be served as a potential targeted dual/multi-drug nanocarrier for the synergistic treatment of nucleoside analogues with hydrophobic antineoplastic drugs.

## 2. Materials and Methods

### 2.1. Materials and Chemicals

5-fluorodeoxyuridine (98%, Alfa Aesar, Shanghai, China), 4,4’-Dimethoxytrityl Chloride (DMT-Cl, 97%, Macklin, Shanghai, China), N, N-Diisopropylethylamine (DIPEA, 97%, Macklin), 2-Cyanoethyl N, N, N′, N′-tetraisopropylphosphordiamidite (95%, Macklin), N-(ε-malemidocaproyloxy) succinimide ester (EMCS, 98%, Aladdin, Shanghai, China), pyrene (95%, Aladdin), curcumin (95%, Aladdin) and 2-(4-amidinophenyl)-6-indolecarbamindine dihydrochloride (DAPI, 95%, Sangon Biotech, Shanghai, China) were used as received. CytoPainter LysoRed Indicator Reagent (ab176828) was bought from Abcam Co., Ltd (Boston, MA, USA). Trypsin, 3-(4,5-dimethylthiazol-2-yl)-2,5-diphenyl-2H-tetrazolium bromide (MTT), phosphate-buffered saline (PBS), Dubelcco’s Modified Eagle’s Medium (DMEM), Roswell Park Memorial Institute-1640 (RPMI-1640), antibiotic-antimycotic (100×) and fetal bovine serum (FBS) were purchased from Wisent Biotechnology (Nanjing) Co. Ltd (Nanjing, China). The caspase 3, 8 and 9 activity detection kits were purchased from BestBio Biotechnology Co., Ltd. (Shanghai, China).

### 2.2. Synthesis of DNA Micelles 

#### 2.2.1. Synthesis of FUdR Phosphoramidite

FUdR phosphoramidite was synthesized according to our previously reported procedure [43,53], and the synthesis route is shown in Figure A1. Pure product (yield 78%) was obtained by silica gel flash chromatography purification (discontinuous gradient of ethyl acetate/methanol (1:0–5:1) with 1 % triethylamine). 1H NMR data of FUdR phosphoramidite follow: 1H NMR (600 MHz, DMSO-d6): δ 11.87 (1H, s), 7.94 (1H, t, J = 6.6 Hz), 7.40 (2H, t, J = 7.2 Hz), 7.32–7.23 (7H, m), 6.88 (4H, J = 8.4 Hz, J = 3.0 Hz), 6.17–6.12 (1H, m), 4.54–4.48 (1H, m), 4.06–3.99 (1H, m), 3.74 (3H, s), 3.73 (3H, s), 3.71–3.59 (2H, m), 3.58–3.53 (1H, m), 3.51–3.46 (1H, m), 3.32–3.28 (1H, m), 3.25–3.19 (1H, m), 2.76 (1H, t, J = 11.4 Hz), 2.64 (1H, t, J = 12 Hz), 2.44–2.39 (1H, m), 2.37–2.27 (1H, m), 1.13 (3H, J = 6.6 Hz), 1.11–1.09 (6H, m), 0.98 (3H, d, J = 6.6 Hz) (Figure A1); 13C NMR (600 MHz, DMSO-d6): δ 158.1, 157.1, 156.9, 148.9, 144.6, 135.3, 135.2, 129.7, 127.8, 127.6, 118.9, 118.7, 113.2, 85.9, 85.8, 84.5, 58.4, 58.2, 55.0, 42.6, 42.5, 24.3, 19.8 (Figure A2).

#### 2.2.2. Solid-Phase Synthesis of cF_10_G_6_-Chl and cF_13_-NH_2_

The affi-F/GQs micelles were composed of cF_10_G_6_-Chl and cF_13_-affibody, which were synthesized following the methods described in [43]. The amphiphilic oligonucleotides were purchased from GenScript. Their nucleotide sequences were described in Table A1.

#### 2.2.3. Synthesis, Purification and Characterization of cF_13_-affibody Conjugate

The amino acid sequence of affibody molecule Z_hcHER2:342_ was MIHHHHHHLQVDNFNKEMRNAYWEIALLPNLNNQQKRAIRSLYDDSQSANLLAEAKKLNDAQAPKVDC.

The cF_13_-affibody conjugate was constructed according to our previous report [38,39], and the synthetic route is described in Figure A2. The obtained DNA-affibody conjugate was analyzed and characterized by UV-vis spectroscopy and 2% agarose gel electrophoresis. The molecular weights of cF_13_-affibody and affibody were also detected by gel permeation chromatography (GPC, Waters 2695, Milford) using polystyrene as standard. The pure cF_13_-affibody was concentrated using Amicon ultracentrifugal filters (MWCO 10 kDa, Merck Millipore, Darmstadt, Germany) and stored at 4 °C.

#### 2.2.4. Self-Assembly of affi-F/GQs

The affi-F/GQs micelles were prepared by mixing the cF_10_G_6_-Chl and cF_13_-affibody (molar ratio = 1:1) in PBS containing 50 mM KCl, and the solution was heated to 75 °C for 15 min, then slowly cooled to room temperature and incubated for 2 h. The obtained DNA micelles were analyzed by 2% agarose gel electrophoresis, and the G-quadruplex in DNA micelles was characterized by CD spectroscopy. 

#### 2.2.5. Preparation of Cur@affi-F/GQs

Cur@affi-F/GQs were prepared by a modified direct dissolution method. Briefly, 5 mg of Cur was dissolved in 1 mL of DMSO, and 100 µL of the above solution was added dropwise to 1 mL of affi-F/GQs (100 µM) aqueous solution. The mixture was heated to 75 °C for 30 min and cooled to room temperature. After 24 h of dark dialysis (dialysis bag with MWCO 10 kDa) at 37 °C, the mixture was centrifuged at 13,000 rpm for 10 min to remove non-encapsulated Cur precipitate, and the supernatant containing Cur@affi-F/GQs was concentrated using Amicon ultracentrifugal filters (MWCO 10 kDa) and stored shielded from light at room temperature.

### 2.3. Characterization

#### 2.3.1. Determination of Critical Micelle Concentration (CMC)

The CMC of affi-F/GQs was determined using pyrene as a probe. The DNA micelle-pyrene mixture was heated to 75 °C for 30 min, then slowly cooled to room temperature and incubated for an additional 24 h. The fluorescence intensity of pyrene was determined by Hitachi F-7000 fluorescence spectrophotometer with an excitation wavelength of 335 nm and emission spectrum of 350–450 nm. The concentration of affi-F/GQs ranged from 20 to 5000 nM, and that of pyrene was 6 × 10^-7^ M. The CMC could be calculated by tracking the intensity ratio of the peak at 373 nm to the peak at 384 nm as a function of affi-F/GQs concentrations.

#### 2.3.2. Stability Analysis of affi-F/GQs

The stability assays of affi-F/GQs were conducted in simulated various in vivo conditions. The affi-F/GQs were incubated in different pH buffers (acidic: pH = 4.5, neutral: pH = 7.4 and alkaline: pH = 9.0) at 37 °C for 2.0 h. Subsequently, DNA micelles were also treated with 0.36 U/mL DNase I for 1.0 h, 2.0 h, 4.0 h and 8.0 h. Finally, they were dissolved into different concentration of BSA (1, 10, 100 and 1000 µM) for 1.0 h. Then, the processed sample was loaded onto 2% agarose gel for electrophoresis, and the gel was stained by GelStain and photographed by Tanon 2500 (Tanon Science & Technology Co., Ltd., Shanghai, China).

#### 2.3.3. Drug Loading Study of Cur@affi-F/GQs

The encapsulation of Cur into Cur@affi-F/GQs was verified by spectral scanning of 230–650 nm using a NanoDrop 2000c spectrophotometer (ThermoFisher Co., Waltham, MA, USA). The encapsulation efficiency (EE) and drug loading (DL) of Cur were also determined by the UV-vis method. The measurement method was as follows: 0.1 mL of Cur@affi-F/GQs solutions (100 µM) were transferred into a 1.0 mL volumetric flask and diluted to the volume with ethanol. The UV-vis spectrophotometer was used to detect the absorbance at 430 nm, and the concentration of Cur was calculated based on the standard curve. The EE and DL of Cur were calculated according to the following formulation:$$EE\left(\%\right)=\frac{weight\;of\;Cur\;in\;Cur@affi-F/GQs}{weight\;of\;total\;Cur}\times 100\%$$



$$DL\left(\%\right)=\frac{weight\;of\;Cur\;in\;Cur@affi-F/GQs}{weight\;of\;Cur@affi-F/GQs}\times 100\%$$



#### 2.3.4. *In Vitro* Drug Release Study of Cur@affi-F/GQs

To determine the *in vitro* release behavior of Cur from DNA micelle, 1 mL Cur@affi-F/GQs solution (100 µM) was added into a dialysis bag with MWCO 10 kDa and then completely submerged into 10 mL release medium (PBS, pH 7.4 or acetate buffer, pH 4.5) and continuously shaken under 100 rpm in an incubator shaker (ZWYR-200D, LABWIT Scientific, Shanghai, China) at 37 °C. At predetermined time intervals, 0.5 mL of release media were removed and replaced with 0.5 mL of fresh release media. 

The *in vitro* nuclease digestion test of Cur@affi-F/GQs was performed to prove the successful release of Cur in cancer cells. DNase II is a mammalian endonuclease with the best activity at acid pH. Therefore, Cur@affi-F/GQs were treated with DNase II (20 U/mL) in acetate buffer (pH 4.5) at 37 °C for different times. The Cur released from Cur@affi-F/GQs was separated using Amicon ultracentrifugal filters (MWCO 3500 Da, Merck Millipore, Darmstadt, Germany).

#### 2.3.5. Circular Dichroism (CD) Spectroscopy

After annealing of the samples, the CD signals of the obtained DNA micelles were measured in the range of 220 nm to 350 nm using Bio-logic MOS-450/SFM300 spectropolarimeter (Bio-Logic Science Instruments, Seyssinet-Pariset, France).

#### 2.3.6. Atomic force microscopy (AFM)

AFM imaging of affi-F/GQs and Cur@affi-F/GQ were performed in tapping mode. 10 µL samples were deposited on mica and adsorb for 20 min. The sample was subsequently washed with ddH_2_O three times, and 40 µL of TAE/Mg^2+^ buffer was added and dried in air at room temperature. The samples were imaged with AFM (5500 AFM/SPM System, Agilent Technologies, Palo Alto, CA, USA).

#### 2.3.7. Transmission Electron Microscopy (TEM) 

The morphology of affi-F/GQs and Cur@affi-F/GQs was examined by TEM (H-600, Hitachi, Ltd., Tokyo, Japan). The affi-F/GQs and Cur@affi-F/GQs were diluted with distilled water and placed on a copper electron microscopy grid and negatively stained with a 2% (*w*/*v*) phosphotungstic acid solution. The excess fluid was removed with a piece of filter paper and then dried in air at room temperature. TEM analysis was done for the dried samples.

#### 2.3.8. Dynamic Light Scattering (DLS)

The DNA micelles were diluted with ultrapure water to 10 µM and then the particle sizes were measured at 37 °C applying Nanobrook Omni (Brookhaven Instruments Corporation, Holtsville, NY, USA).

### 2.4. Cellular Experiments

#### 2.4.1. Cell Lines and Cell Cultures

Human gastric cancer cell line N87 and MGC 803 cells were obtained from the cell resource center at the Shanghai Biological Sciences Institute (Chinese Academy of Sciences, Shanghai, China). N87 cells were cultured in RPMI-1640 with FBS and penicillin-streptomycin solution at the concentration of 10% and 1%, respectively. MGC 803 cells were cultured in high-glucose DMEM containing 10% FBS and 1% penicillin-streptomycin solution. The cells were placed at 37 °C in a humidified atmosphere containing 5% CO_2_.

#### 2.4.2. Cellular Uptake

The uptake of DNA micelles by gastric cancer was carried on N87 and MGC 803 cells, which were obtained from the cell resource center at the Shanghai Biological Sciences Institute (Chinese Academy of Sciences, Shanghai, China). For fluorescent imaging, N87 and MGC 803 cells were separately inoculated onto the laser confocal dish (NEST, Wuxi, China) at a density of 1 × 10^5^ cells per well and cultured overnight at 37 °C. Then, the medium was replaced with fresh medium containing FAM-labeled F/GQs and affi-F/GQs (terminal concentration of FAM, 5 µM). After 1 h of incubation, the culture solution was removed and the cells were washed twice with PBS and the cells were treated with 4% paraformaldehyde fix solution for 20 min, and the nucleus was stained using 2.5 µg/mL DAPI for 30 min. Finally, fluorescence images were taken of the samples via a confocal laser scanning microscopy (CLSM, Zeiss LSM 810, Oberkochen, Germany).

For colocalization with lysosomes, N87 cells were stained with LysoTracker Red for 0.5 h after 1 h of incubation with affi-F/GQs at 37 °C and then fixed with paraformaldehyde fix solution for 15 min. Fluorescent images were captured by a fluorescence microscopy with an excitation 577 nm, and the Pearson’s correlation was analyzed by image analysis software (Image Pro-Plus). All images were acquired using a 40× oil objective.

Cellular uptake behavior of Cur@affi-F/GQs was investigated after incubation with N87 cells for 1.0, 2.0 and 4.0 h. N87 cells (1 × 10^5^ cells) were placed into laser confocal dishes and cultured overnight, then treated with Cur@affi-F/GQs at a final Cur concentration of 10 µM. At definite time intervals, the media were removed and cautiously washed with PBS before being fixed with 4% paraformaldehyde for 20 min. The cell nucleus was counterstained by DAPI and imaged on fluorescence microscopy with an excitation of 405 nm procedure, and the Cur@affi-F/GQs treated cells in different dishes were observed at an excitation of 488 nm according to standard. All images were acquired using a 40× oil objective.

#### 2.4.3. *In Vitro* Cytotoxicity

Firstly, the suspensions of N87 and MGC 803 cells were, respectively, seeded in 96-well plates (5 × 10^3^ cells per well) and cultured at 37 °C for 24 h. Then, 100 µL of fresh medium containing different concentrations of FUdR, F/GQs, affi-F/GQs, Cur, FUdR/Cur (6:1) and Cur@affi-F/GQs were added into each well. After incubation for 48 h, 10 µL of MTT (5 mg/mL) was added into each well, and the plates were incubated at 37 °C for 4 h. The medium was carefully removed and replaced by 100 µL DMSO. The absorbance at 570 nm was measured by a microplate reader. Cell viability was calculated by the formula below:$$Cell\;viability\left(\%\right)=\frac{OD_{570\left(Sample\right)}-OD_{570\left(Blank\right)}}{OD_{570\left(Control\right)}-OD_{570\left(Blank\right)}}\times 100\%$$

In addition, the online calculator (https://www.aatbio.com/tools/ic50-calculator) was used to determine the IC50 value of FUdR and Cur alone and in combination.

The combination index (CI) was calculated according to the following formula:$$CI=\frac{D_{FUdR}-D_{Cur}}{{\left(D_{x}\right)}_{FUDR}-{\left(D_{x}\right)}_{Cur}}$$(D_x_)_FUdR_ and (D_x_)_Cur_ are the concentrations of FUdR and Cur as a single drug with a certain inhibition rate (e.g., IC50). D_FudR_ and D_Cur_ are the concentrations of FUdR and Cur in combination at the given inhibition rate, respectively. The molar ratio of FUdR and Cur in FUdR/Cur (6:1) and Cur@affi-F/GQs was fixed at 6:1. CI > 1, = 1 and < 1 represented antagonism, additive and synergism, respectively.

#### 2.4.4. Western Blot Analysis

Approximately 2 × 10^5^ of N87 and MGC 803 cells were seeded on 6-well plates. After 24 h of incubation at 37 °C and 5% CO_2_, the cells were treated with FUdR, F/GQs, affi-F/GQs, Cur, FUdR/Cur (6:1) and Cur@affi-F/GQs (containing FUdR, 60 µM or Cur, 10µM), respectively. After measuring the protein concentration, the protein was resolved on 12% SDS-PAGE and transferred to the polyvinylidene difluoride (PVDF) membranes. The membranes were incubated with primary antibodies against Bcl-2, Bax and β-actin (bsm-33047M, bsm-33283M and bsm-33036M, Bioss Antibodies) overnight at 4 °C and then incubated with secondary antibody-conjugated horseradish peroxidase for 1 h, followed by washing with TBST buffer two times. Finally, the blots were detected with the Tanon 5200 ECL system (Tanon Science & Technology Co., Ltd., Shanghai, China) and analyzed by Image J software (Image J 2.1.4.7, Wayne Rashand, National Institutes of Health, Bethesda, MD, USA).

#### 2.4.5. Caspase Activity Assays

The cells’ treatment and grouping information were the same as for the western blotting experiment. After being treated for 10 h (caspase 8, caspase 9 tests) or 24 h (caspase 3 test), the cancer cells were harvested and lysed for 15 min on ice following the kit instructions. The supernatants were collected by centrifugation (18,000 rpm at 4 °C for 10 min), and the protein concentration was determined by the BCA method. The substrates were added to the supernatant (final concentration, 2 mg/mL) and incubated at 37 °C for 4 h. The absorbance was measured at a wavelength of 405 nm on Synergy HT microplate reader (Biotek Instruments, Inc., Green Mountains, VT, USA). Finally, the relative caspase activity was calculated according to the ratio of the absorbance value of apoptosis-induced cells to that of blank control cells.

### 2.5. Statistical Analysis

All the experiments samples were repeated at least three times unless otherwise stated. The results are expressed as mean ± standard deviation (SD). The statistical differences among groups were determined by Newman–Keuls analysis. A value of * *p* < 0.05 was considered significant, ** *p* < 0.01 was considered highly significant and *** *p* < 0.001 was considered extremely significant.

## 3. Results and discussion

### 3.1. Structural Design of affi-F/GQs

As shown in Figure 1A,B, affi-F/GQs were composed of cholesterol-modified affibody-FUdR-DNA hybrid strands. Each hybrid strand consisted of a cholesterol-modified FUdR-DNA strand (cF_10_G_6_-Chl) and an affibody-modified FUdR-DNA strand (cF_13_-affibody). cF_10_G_6_-Chl, an amphiphilic lipid-DNA strand, was responsible for the formation of DNA micelles, including two segments: 29-mer FUdR-DNA strands as corona and cholesterol tails as core. From its 5’ end to 3’ end, the 29-mer FUdR-DNA strand was divided into a complementary region (13 mer), FUdR drug-loading region (10 consecutive FUdRs) and G-rich region (6 consecutive Gs). The cholesterol tail was covalently linked to the 3’ end of DNA strand by solid-phase synthesis. In addition, the cF_13_-affibody was a 26-mer FUdR-DNA strand with a target molecule affibody, which also contained a complementary region (13 mer) and drug-loading region (13 consecutive FUdRs). The above two DNA conjugates, cF_10_G_6_-Chl and cF_13_-affibody, formed a hybrid strand by base pairing in the complementary regions, and each hybrid strand contained 23 anti-cancer FUdRs. In aqueous solution, each unimer of affi-F/GQs was self-assembled by four DNA hybrid strands through Hoogsteen hydrogen bond and π-π stacking between G-quartets in the G-rich region and the hydrophobic interaction of cholesterol. In the DNA micelles, the G-quadruplexes were in the innermost layer of the hydrophilic shell and adjacent to the hydrophobic core, which acted as an adhesive to stabilize the structure of DNA micelle. FUdRs were covalently integrated into the DNA strands and located in the middle layer of the hydrophilic shell, which helped to achieve the sustained release of drugs. Affibody molecules were decorated on the surface of DNA micelle, which enabled affi-F/GQs to enter cancer cells through receptor-mediated endocytosis and produced anti-tumor effects.

### 3.2. Synthesis and Characterization of cF_10_G_6_-Chl and cF_13_-affibody

Based on the above design, two DNA strands containing FUdR were synthesized and named cF_10_G_6_-Chl and cF_13_-NH_2_ (Sequence information was listed in Table A1). To integrate the nucleoside antitumor drug FUdR into the DNA strand, FUdR was derivatized into its phosphamide monomer by two steps of chemical modification (Figure A1). The monomers were used to prepare the two FUdR-DNA strands by solid-phase synthesis with the phosphoramidite monomers of A, T, G and C. Then, cholesterol as a hydrophobic block was covalently linked to the 3′ end of cF_10_G_6_-Chl by DNA solid phase synthesis. In addition, the 3′ end of cF_13_-NH_2_ was modified with amino group, which was used to connect the targeting affibody molecule. The two FUdR-DNA strands were characterized by mass spectrometry, and the results suggested that the actual molecular weights of cF_10_G_6_-Chl and cF_13_-NH_2_ were substantially the same as their theoretical molecular weights (Figure 1A). The results of denatured polyacrylamide gel electrophoresis (Figure 1B) also confirmed that cF_10_G_6_-Chl and cF_13_-NH_2_ were successfully synthesized and that their purity was more than 98%.

Next, the targeting affibody molecule, Z_hcHER2:342_, was linked to the 3' end of cF_13_-NH_2_ through chemical synthesis. Z_hcHER2:342_ was constructed based on Z_HER2:342_, and its N-terminal contained His-Tag, which facilitated the purification of Z_hcHER2:342_ by affinity chromatography (Figure A3). Z_hcHER2:342_ and cF_13_-NH_2_ was effectively conjugated through a EMCS linker according to Figure A2. The coupled product, cF_13_-affibody, was purified via two steps (Capto DEAE column chromatography and HisTrap HP column chromatography) to remove unreacted cF_13_-NH_2_ and excessive affibody molecules. UV-vis analysis showed that the absorption peak of cF_13_-affibody had a red shift of about 10 nm compared with that of cF_13_-NH_2_, and its absorption spectrum was also significantly different from that of cF_13_-NH_2_ and affibody molecule (Figure 1C). Moreover, agarose gel electrophoresis indicated that cF_13_-affibody had a slower migration rate than that of cF_13_-NH_2_, (Figure 1D). GPC analysis also indicated that the molecular weights were 8.3 and 16.6 kDa for affibody and cF_13_-affibody, respectively (Figure A4). All the above results proved that cF_13_-NH_2_ and affibody were connected to form cF_13_-affibody.

### 3.3. Self-assembly and Characterization of affi-F/GQs

The synthesized cF_10_G_6_-Chl and cF_13_-affibody were mixed in equal molar ratio in the reaction buffer, and affi-F/GQs were formed by self-assembly after simple heating-cooling treatment. The structure of affi-F/GQs was identified by 2% agarose gel electrophoresis. It can be seen from Figure 2A that the electrophoresis band of cF_10_G_6_ (without cholesterol, Lane 1) was located at the front of the gel, suggesting that there was no formation of G-quadruplex structure. The reason might be that the extension of oligonucleotide linked to six consecutive guanines (Gs) increased the spatial barrier and made it difficult to form G-quadruplex. However, the amphiphilic cF_10_G_6_-Chl can easily self-assemble into G-quadruplex by virtue of the hydrophobicity of cholesterol [20], which led to the enlargement of nanoparticles and the decrease of mobility (Figure 2A, Lane 2). When six consecutive Gs in the cF_10_G_6_-Chl were replaced with six consecutive thymines (Ts), the DNA strand cF_10_T_6_-Chl (Figure 2A, Lane 3) still had fast mobility, indicating that even if the DNA strand was linked with cholesterol, cF_10_T_6_-Chl did not self-assemble into a quadruplex structure, because the six consecutive Ts could not form a Hoogsteen hydrogen bond and π-π stacking. In addition, affi-F/GQs (Figure 2A, Lane 4) also exhibited slow mobility due to the formation of correct G-quadruplex structure. It was also proved that the complementary pairing process between cF_13_-affibody and cF_10_G_6_-Chl did not hinder the formation of G-quadruplex structure. Then, the structure of G-quadruplex in DNA micelles was further confirmed by circular dichroism (CD). In the CD spectra (Figure 2B), both positive bands at 260 nm and negative bands at 240 nm were attributed to the G-quadruplex structure of cF_10_G_6_-Chl and affi-F/GQs, which were the characteristic bands of tetramolecular parallel G-quadruplex structure [54,55]. As a control, cF_10_T_6_-Chl was also analyzed by CD. The positive band at 280 nm and the negative band at 250 nm were displayed, and no characteristic band of G-quadruplex was observed. It was thus inferred that no G-quadruplex structure was formed in cF_10_T_6_-Chl. Subsequently, affi-F/GQs were characterized by dynamic light scattering (DLS). The results showed that the hydrodynamic size of affi-F/GQs was 108.02 ± 7.69 nm and that the polydispersity index (PDI) was 0.118 ± 0.033 (Figure 2C). Furthermore, atomic force microscopy (AFM) and transmission electron microscope (TEM) analysis indicated their spherical structure and uniform size (about 90 nm) (Figure 2D, Figure A5). 

### 3.4. Determination of Critical Micelle Concentration

CMC is an important parameter for the analysis of micelle formation, and it is commonly detected with pyrene fluorescence probe [56]. There are five peaks in the fluorescence spectrum of pyrene in the range of 350–500 nm (Figure A6). The ratio of fluorescence intensity of the first peak (373 nm) to that of the third peak (384 nm) shows a cross point as the micelle concentration approaches the CMC. When affi-F/GQs were dispersed in aqueous solution containing pyrene, the concentration of affi-F/GQs at the cross point of the I373/384 ratio was approximately 200 nM (Figure 3A), which was the CMC value of affi-F/GQs. To ensure the formation of DNA micelles, the affi-F/GQs involved in this study were assembled at concentrations of more than 200 nM.

### 3.5. Stability Analysis of affi-F/GQs

To evaluate the stability, affi-F/GQs were incubated in solution with a different pH, DNase I treatment time and BSA concentrations, respectively. The processed samples were analyzed by 2% agarose gel electrophoresis. As shown in Figure 3B, affi-F/GQs still maintained stable micelle structure at neutral and alkaline conditions (pH 7.4 and 9.0), but there is a little dissociation detected from the lower band at acidic condition (pH 4.5), which may be due to the destruction of the Hoogsteen hydrogen bond by acidic pH. In addition, affi-F/GQs were incubated with DNase I at 0.36 U/mL (plasma level) for 8 h and did not show much degradation, implying their good nuclease tolerance. The stability of affi-F/GQs in BSA solution was also investigated. It was found that most of the affi-F/GQs remained intact with the increasing concentration of BSA from 1 to 1000 µM, and only a small number of affi-F/GQs were dissociated and bound to BSA at high concentration (Figure 3B). These results show that G-quadruplex-locked DNA micelles had excellent stability and effectively prevented the destructive effect of serum albumin on their structures, which is not possessed by most polymeric micelles [57].

### 3.6. Preparation and Characterization of Cur@affi-F/GQs

According to Figure 1C, Cur was encapsulated into the hydrophobic core of affi-F/GQs by modified direct dissolution [58] to form dual drug-loaded Cur@affi-F/GQs. The encapsulation efficiency (EE) and drug loading (DL) of Cur in Cur@affi-F/GQs were analyzed by UV-Vis spectrophotometer. Compared with the absorption spectrum of affi-F/GQs, the characteristic absorption peaks of Cur were observed in the spectrum of Cur@affi-F/GQs (Figure 4A), which proved that the Cur molecules were loaded in affi-F/GQs. According to the standard curve of Cur, 10 µM Cur@affi-F/GQs contained 38.8 µM Cur (Figure 4B), and the EE and DL of Cur in Cur@affi-F/GQs were calculated to be 28.6% and 5.6%, respectively. Based on the design of affi-F/GQs, 10 µM Cur@affi-F/GQs contained 230 µM FUdR, and the DL of FUdR in Cur@affi-F/GQs was 21.1% in weight, which was more precise and controllable than the physical embedding of other 5-fluorouracil-loading nanoparticle [59,60,61]. Therefore, the mole ratio of FUdR to Cur in Cur@affi-F/GQs was fixed at approximately 6:1. The average hydrodynamic diameter of Cur@affi-F/GQs was 132.20 ± 18.92 nm measured by DLS, which was bigger than that of affi-F/GQs (108.02 ± 7.69 nm). The PDI of Cur@affi-F/GQs was 0.288 ± 0.022 (Figure 4C). The morphology of the Cur@affi-F/GQs, determined by AFM and TEM, was shown in Figure 4D and Figure A5, and this image revealed that Cur@affi-F/GQs were more regular, larger and denser spheres compared with affi-F/GQs.

### 3.7. In Vitro Drug Release Behavior

The *in vitro* release behavior of Cur@affi-F/GQ was investigated using a dialysis method. As displayed in Figure 5A, a sustained release was found either in the simulated physiological conditions (pH 7.4) or lysosome acidic conditions (pH 4.5), and the cumulative release of Cur reached 43.6% and 66.7%, respectively, after 48 h of dialysis. In addition, it was noticed that the initial rapid release of Cur occurred at both pH conditions, which might be attributed to the difference in drug concentration between the DNA micelles and the external solution. In its late release stage, Cur was slowly released from the lipid core of DNA micelles to the external solution through diffusion, thus playing a sustained release effect. Furthermore, the low cumulative release of Cur@affi-F/GQs at pH 7.4 indicated that it effectively reduced the leakage of Cur and allowed the drug to be delivered to the tumor site as much as possible. Conversely, the high cumulative release at pH 4.5 facilitated the drug’s efficacy in tumor cells. Notably, no free FUdR was detected in the above process, which indicated that FUdRs integrated into Cur@affi-F/GQs by phospholipid bonds was more stable than physical encapsulation [62,63] and could effectively avoid drug leakage during blood transport.

Subsequently, the nuclease hydrolysis of Cur@affi-F/GQs was evaluated *in vitro* to explore the drug release behavior in cancer cells. DNase II, as a nuclease, is widely distributed in the lysosomes of animal cells, which is responsible for the degradation of exogenous DNA. After Cur@affi-F/GQs were incubated with DNase II at pH 4.5 for 2 h, the micelle structure was almost destroyed completely (Figure 5B) and the release rate of Cur reached 83.1% (Figure 5C). Then, the dissociated DNA strand was completely degraded after incubation with DNase II for 12 h (Figure 5B). Thus, the FUdRs integrated into DNA strands were released and combined with Cur to kill cancer cells.

### 3.8. Cellular Uptake

The cellular uptake of affi-F/GQs or F/GQs (composed of cF_13_-NH_2_ and cF_10_G_6_-Chl) by gastric cancer N87 (HER2 positive cell) and MGC 803 cells (HER2 negative cell) was monitored through CLSM to investigate the targeting property of affi-F/GQ as drug carrier. As illustrated in Figure 6A, compared with N87 cells treated with FAM labeled F/GQs, the stronger green fluorescence signals were detected in their treatment with affi-F/GQs. Conversely, the fluorescence intensity of MGC 803 cells treated with affi-F/GQs was lower than that of MGC 803 cells treated with F/GQs. The results indicated that affibody distributed in the shell of affi-F/GQs greatly increased the specific binding ability to HER2 positive cancer cells, resulting in a large uptake of affi-F/GQs. Conversely, affi-F/GQs bound weakly to HER2 negative cancer cells due to the lack of receptors, which may effectively avoid the high intake and reduce the side effects of chemotherapy drugs. 

The cellular localization of affi-F/GQs was further investigated by CLSM. It was found from Figure 6B that green fluorescence from affi-F/GQs was localized in the lysosomes of N87cells by taking lysosome staining with LysoTracker Red. Then, Image Pro-Plus was used to calculate Pearson’s correlation coefficient (r) for analyzing the co-localization level. The r value was 0.73, which indicated that most affi-F/GQs were trapped in lysosomal compartments, further confirming that affi-F/GQs enter HER2 positive cancer cells through receptor-mediated endocytosis.

Subsequently, it was investigated that Cur@affi-F/GQs were absorbed by HER2 positive gastric cancer cells. In this process, the intrinsic green fluorescence of Cur was used for imaging analysis. The image of N87 cells incubated with Cur@affi-F/GQs (Figure 6C) showed that the green fluorescence of Cur aggregated into dots in the cells within 1 h, and with the incubation time extended to 4 h, the green fluorescence almost covered the whole cell. It was inferred from the above phenomenon that Cur@affi-F/GQs first existed in the lysosome as intact micelles after entering the cells through endocytosis and that they then rapidly dissociated under the action of lysosomal acid environment and DNase II. Finally, Cur was gradually released and distributed in the whole cell.

### 3.9. Cytotoxicity and Synergistic Effect Studies

Gastric cancer is the second leading cause of cancer-related mortality worldwide [64], and chemotherapy, such as 5-FU [65] or FOLFOX (0.1 µM 5-FU and 5 µM oxaliplatin) [66], remains the standard care treatment approach for patients with advanced-stage disease and after surgery. However, drug resistance and toxicity induced by 5-FU and inadequate targeting were the main obstacles for the effective treatment of patients with gastric cancer. Thus, Cur was co-loaded with FUdR to sensitize gastric cancer cells and reduce drug resistance, and affibodies were introduced to provide targeting for chemotherapeutic drug delivery in our study.

The selective antitumor effects of affi-F/GQs, F/GQs and free FUdR on HER2 positive /negative gastric cancer cells were detected by the MTT method. A concentration-dependent increase in cytotoxicity was present in N87 and MGC 803 cells after treatment with free FUdR, F/GQs and affi-F/GQs (Figure 7A). The affi-F/GQs were of higher cell cytotoxicity in N87 cells, compared with F/GQs and free FUdR. The IC 50 value of affi-F/GQs (23.2 μM) was lower than that of nontargeting F/GQs (37.2 μM) and free FUdR (55.2 μM) (Figure 7C). This further suggested that the higher cytotoxicity resulted from the improved targeting efficacy of affi-F/GQs. However, the cytotoxicity of affi-F/GQs to MGC 803 cells was lower than that of F/GQs, and the IC50 value was 43.8 μM, which was higher than that of F/GQs (32.6 μM). This may be due to the reduced uptake of affi-F/GQs by cancer cells with low HER2 expression level. These results also support the view that target molecule affibody and DNA micelle structure enhanced the antitumor effect of FUdR.

Furthermore, the inhibitory and the synergistic effect of Cur@affi-F/GQs on N87 cells were evaluated. Compared with FUdR and Cur alone, both FUdR/Cur (6:1) (a physical mixture of FUdR and Cur at a molar ratio of 6:1) and Cur@affi-F/GQs exhibited higher cell cytotoxicity against N87 cells (Figure 7B), especially Cur@affi-F/GQs, in which the IC50 values of FUdR and Cur were 13.1 μM and 2.2 μM, respectively, which was the lowest in all treatment groups (Figure 7C). The synergetic effect of FUdR and Cur was quantified through the analysis of combination index (CI) as previously described [67]. CI > 1, =1 or <1 were applied to represent antagonism, additivity and synergy, respectively. According to Figure 7D, the CI value of Cur@affi-F/GQs was lower than that of physical combination (0.417 vs. 0.993), which indicated a remarkable synergistic anticancer effect of FUdR and Cur in Cur@affi-F/GQs. The reason for these results may be related to the increase in the sensitivities of cancer cells to FUdR caused by Cur, as well as the targeted DNA-micellar drug delivery platform.

### 3.10. Exploration of Anticancer Mechanism

Previous reports have confirmed that chemotherapy with FUdR can lead cancer cells to enter the apoptotic pathway via the mitochondrial pathway regulated by the Bcl-2 protein family [68]. The regulation of Bcl-2 and Bax expression has a significant effect on the release of cytochrome c and the formation of apoptosome [69]. Beyond that, caspases protease family are the key enzymes to induce apoptosis, and the activities of caspase 3, 8 and 9 are regulated by various factors, including the Bcl-2 protein family [70]. Therefore, the down-regulation of Bcl-2, up-regulation of Bax and the enhancement of caspase 3, 8 and 9 activity can increase the apoptosis rate of tumor cells.

In order to reveal the anticancer mechanism of DNA micelles, the apoptosis-related proteins were detected by western blot and caspase activity assay kits in our study. As shown in Figure 8A, affi-F/GQs significantly down-regulated Bcl-2 expression and up-regulated Bax expression in N87 cells, compared to the control group (PBS) or the groups treated with free FUdR or F/GQs, but the expression changes of Bcl-2 and Bax caused by affi-F/GQs were comparable to those of F/GQs in MGC803 cells. The Bax/Bcl-2 ratio was 1.25 in N87 cells treated with affi-F/GQs, which was about two-fold higher than that of F/GQs (0.62) (Figure 8C). Moreover, the results of caspase activity (Figure 8D,E) indicated that there was higher caspase 3, 8 and 9 activities in N87 cells than in MGC803 cells after treated with affi-F/GQs. All findings clarified the good selectivity and cytotoxicity of affi-F/GQs on HER2 positive gastric cancer cells.

In addition, the effects of FUdR and Cur alone and their combination (FUdR/Cur (6:1), Cur@F/GQs and Cur@affi-F/GQs) on the expression of Bcl-2 and Bax in N87 cells were also analyzed. According to Figure 8B, the expression levels of Bcl-2 and Bax were markedly changed after N87 cells were treated with FUdR/Cur (6:1), Cur@F/GQs and Cur@affi-F/GQs, respectively. It can be seen from Figure 8C that the Bax/Bcl-2 ratio in N87 cells treated with Cur@affi-F/GQs was 1.83, which was higher than that of other treatment groups. Furthermore, the highest activity of caspase 3, 8 and 9 was noticed when the cells were treated with Cur@affi-F/GQs, and their activities were 14.6, 8.7 and 10.5 times of that of the control group (Figure 8F). Based on the above data, it can be inferred that Cur improved the activity of apoptosis proteins and sensitized gastric cancer cells to FUdR and that the drug carrier affi-F/GQs played a significant role in promoting the synergistic effect of FUdR and Cur.

## 4. Conclusions

In the present study, the G-quadruplex DNA micelle with covalently loaded FUdRs and targeting affibody was first synthesized, a synthesis that exhibited excellent stability and maintained an intact micellar structure in various simulated environments. Furthermore, affi-F/GQs showed specific binding ability and selective inhibition on HER2 positive gastric cancer cells. Based on the micelle characteristic of affi-F/GQs, Cur was wrapped in the hydrophobic core of DNA micelles to realize the co-loading of FUdR and Cur. The *in vitro* tests suggested that Cur@affi-F/GQs had the strongest antitumor activity and the best synergistic effect on N87 cells. The anticancer mechanism studies confirmed that Cur@affi-F/GQs significantly enhanced FUdR-induced Bcl-2/Bax-caspase 8,9-caspase 3 apoptosis pathway. Therefore, the affibody-G-quadruplex micelles can be used as a drug delivery platform for the simultaneous transport of nucleoside antitumor drugs and hydrophobic anticancer drugs. 

## Data Availability

Not applicable.

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
