# Peer review of "Affibody Modified G-quadruplex DNA Micelles Incorporating Polymeric 5-Fluorodeoxyuridine for Targeted Delivery of Curcumin to Enhance Synergetic Therapy of HER2 Positive Gastric Cancer"

_nanomaterials, 2022, doi:10.3390/nano12040696_

Round 1

Reviewer 1 Report

In this paper, a new Affibody Modified G-quadruplex DNA Micelles Incorporating  

Polymeric 5-Fluorodeoxyuridine was synthesized for targeted delivery of Curcumin to Enhance Synergetic Therapy of HER2 Positive Gastric Cancer. Followings are my specific queries:

  1. There are some results that show the affibody attached to DNA strand and protein but what evidences show that the G-quadruplex is formed?
  2. The authors mentioned that their method is expected to be a promising treatment strategy for HER2 positive cancer (p. 17, lines 593-594). They should add the evidences that show their method can be used for cancer treatment or delete this sentence.
  3. The authors must compare their results with the previous results and mention the advantages and disadvantages of their method comparing the previous ones.
  4. What is the advantages and disadvantages of this method comparing the current Gastric cancer treatment methods that are already using?
  5. I recommend the authors add some other related references to their manuscript for improving the paper quality. I recommend the following references that are very matched to your study:

[1] J. Bang, H. Park, W.I. Choi, D. Sung, J.H. Lee, K.Y. Lee, S. Kim, Sensitive detection of dengue virus NS1 by highly stable affibody-functionalized gold nanoparticles, New Journal of Chemistry, 42 (2018) 12607-12614.

[2] H. Ilkhani, A. Ravalli, G. Marrazza, Design of an Affibody-Based Recognition Strategy for Human Epidermal Growth Factor Receptor 2 (HER2) Detection by Electrochemical Biosensors, Chemosensors, 4 (2016) 23.

[3] J. Liu, D. Cui, Y. Jiang, Y. Li, Z. Liu, L. Tao, Q. Zhao, A. Diao, Selection and characterization of a novel affibody peptide and its application in a two-site ELISA for the detection of cancer biomarker alpha-fetoprotein, International journal of biological macromolecules, 166 (2021) 884-892.

[4] H. Ilkhani, T. Hughes, J. Li, C.J. Zhong, M. Hepel, Nanostructured SERS-electrochemical biosensors for testing of anticancer drug interactions with DNA, Biosensors and Bioelectronics, 80 (2016) 257-264.

[5] H. Ilkhani, C.-J. Zhong, M. Hepel, Magneto-Plasmonic Nanoparticle Grid Biosensor with Enhanced Raman Scattering and Electrochemical Transduction for the Development of Nanocarriers for Targeted Delivery of Protected Anticancer Drugs, Nanomaterials, 11 (2021) 1326.

[6] H. Ilkhani, M. Sarparast, A. Noori, S.Z. Bathaie, M.F. Mousavi, Electrochemical aptamer/antibody based sandwich immunosensor for the detection of EGFR, a cancer biomarker, using gold nanoparticles as a signaling probe, Biosensors and Bioelectronics, 74 (2015) 491-497.

Author Response

Dear Reviewer:

Thank you for your advice.

Our manuscript has been revised as requested and submitted for your further consideration. All the questions and issues in comments are answered and highlighted in yellow color in the manuscript.  Please see the attachment!

Reviewer 2 Report

The paper reported an affibody modified G-quadruplex DNA micellar prodrug (affi-F/GQs) of hydrophilic 5-fluorodeoxyuridine (FUdR) by integrating polymeric FUdRs into DNA strands. curcumin (Cur) is co-loaded into affi-F/GQs micelles to prepare the dual drug-loaded DNA micelles (Cur@affi-F/GQs). The formulations were well characterized and evaluated their application with appropriate results and assays. The paper can be accepted after considering the following results.

  1. The EE% was calculated to be 14.1% which is too low in comparison to other studies. Please explain what is the importance of your work?
  2. Introduction can be improved by discussing some similar studies in this field. The authors can use: https://doi.org/10.1021/acsomega.1c03816, 10.3390/nano11030817, https://doi.org/10.1016/j.molliq.2021.118271
  3. How the samples were prepared for TEM analysis? Detailed methodology is expected.
  4. Some abbreviations were used without explanation. Please check and correct
  5. Figures 4, and scheme 1 have low resolution. Please enhance
  6. The authors mentioned that EE%, DL%, and release behavior were determined with a UV-Vis spectrophotometer. Other compounds of your formulations didn’t show interfering peaks at 430 nm area? Is UV-Vis a valid test for EE assay with this low concentration? What was the LOD of the instrument for your drug? Why don’t use HPLC as many authors use this technique? Please clarify.

Author Response

Response to Reviewer 2 Comments

Point 1: The EE% was calculated to be 14.1% which is too low in comparison to other studies. Please explain what is the importance of your work?

Response 1: Our work aims to provide a novel strategy for the co-delivery of hydrophilic/ hydrophobic drugs, especially the loading of hydrophilic drugs. It is difficult to precisely control the formulation by encapsulating hydrophilic drug molecules inside nanoparticles, and effectively prevent drug leakage during blood circulation. Curcumin, as a hydrophobic drug, could sensitize gastric cancer cells well at lower concentrations (IC50 = 2.2µM), resulting in no optimization of encapsulating conditions. At present, the optimized EE% of curcumin has reached 28.6%, and the optimized method and results were added in lines 173-180, 453-455 of the manuscript and highlighted in yellow.

In addition, curcumin is being designed and modified to facilitate its covalent attachment to DNA strands for replacing cholesterol to form a hydrophobic core.

Point 2: Introduction can be improved by discussing some similar studies in this field. The authors can use: https://doi.org/10.1021/acsomega.1c03816, https://doi.org/10.3390/nano11030817, https://doi.org/10.1016/j.molliq.2021.118271

Response 2: Thanks for the reviewer's suggestion and the related references have been added to the Introduction in our manuscript (references 12, 13, 50) and highlighted in yellow.

Point 3: How the samples were prepared for TEM analysis? Detailed methodology is expected.

Response 3: The detailed TEM methodology was added to ‘2. Materials and Methods, 2.3.7’(lines 232-238).

Point 4: Some abbreviations were used without explanation. Please check and correct

Response 4: The abbreviations without explanation have been corrected and highlighted in yellow.

Point 5: Figures 4, and scheme 1 have low resolution. Please enhance

Response 4: Figures 4, and Scheme 1 have been revised and highlighted in yellow.

Point 6: The authors mentioned that EE%, DL%, and release behavior were determined with a UV-Vis spectrophotometer. Other compounds of your formulations didn’t show interfering peaks at 430 nm area? Is UV-Vis a valid test for EE assay with this low concentration? What was the LOD of the instrument for your drug? Why don’t use HPLC as many authors use this technique? Please clarify.

Response 6: In our nanoparticle, other compounds are mainly DNA and proteins (affibody), and their absorption spectra are shown in Figure 4A, with absorption peaks at 230-300 nm, so they will not interfere with the determination of curcumin.

From the standard curve (Figure 4B), it can be seen that curcumin has very good linearity when the concentration is between 3-48 µM, and curcumin concentration measured in 10 µM Cur@affi-F/GQs was 38.8 µM.

The LOD of the instrument for curcumin was 0.85 µM, and the measurement and calculation methods are as follows: LOD was experimentally verified by diluting the known concentration of Curcumin until the average responses were approximately 3 times the standard deviation of the responses for six replicate determinations. LOD was calculated according to the formulae: LOD was calculated according to the formulae: LOD = 3.3 S/M (where S is the standard deviation of the absorbance of the sample and M is the slope of the calibrations curve). The relevant literature is as follows:

  1. Kadam P.V., Bhingare C.L., Nikam R. Y., Pawar S. A., Development and validation of UV Spectrophotometric method for the estimation of Curcumin in cream formulation, Pharmaceutical Methods, 2013, 4, 43-45. https://doi.org/10.1016/j.phme.2013.08.002.
  2. Sharma K.A., Agarwal S.S., Gupta M. Development and validation of UV Spectrophotometric method for the estimation of Curcumin in bulk drug and pharmaceutical dosage forms. Int.J. Drug Dev. 2012, 4, 375-379.

When the content determination of curcumin in DNA micelles is performed by HPLC, additional extraction operations are required, which will result in the loss of the drug, thus making the results of EE%, DL% inaccurate. The UV-Vis test is more rapid and accurate, and some literature has reported, listed as follows:

  1. Hiremath, C. G., Kariduraganavar, M. Y., & Hiremath, M. B. (2018). Synergistic delivery of 5-fluorouracil and curcumin using human serum albumin-coated iron oxide nanoparticles by folic acid targeting. Progress in biomaterials, 7(4), 297–306. https://doi.org/10.1007/s40204-018-0104-3
  2. Rahdar A.; Hajinezhad M.R.; Sargazi S.; Zaboli M.; Barani M.; Baino F.; Bilal M.; Sanchooli E. Biochemical, ameliorative and cytotoxic effects of newly synthesized curcumin microemulsions: evidence from in vitro and in vivo studies. Nanomaterials (Basel) 2021, 11, 817. https://doi.org/10.3390/nano11030817
  3. Balasubramanian S, Girija AR, Nagaoka Y, et al. Curcumin and 5-fluorouracil-loaded, folate- and transferrin-decorated polymeric magnetic nanoformulation: a synergistic cancer therapeutic approach, accelerated by magnetic hyperthermia. International Journal of Nanomedicine. 2014 ; 9:437-459.https://doi.org/10.2147/ijn.s49882

Please see the attachment for the revised manuscript

Round 2

Reviewer 2 Report

The authors addressed all my concerns and the paper can be accepted in the current format.